# Rethinking the Effect of Uninformative Class Name in Prompt Learning

Fengmao Lv
Southwest Jiaotong University
Chengdu, China
fengmaolv@swjtu.edu.cn

Changru Nie
Southwest Jiaotong University
Chengdu, China
niechangru@my.swjtu.edu.cn

Jianyang Zhang*
University of Electronic Science and
Technology of China
Chengdu, China
jianyangzhang@std.uestc.edu.cn

Guowu Yang
University of Electronic Science and
Technology of China
Chengdu, China
guowu@uestc.edu.cn

Guosheng Lin
Nanyang Technological University
Singapore, Singapore
gslin@ntu.edu.sg

Xiao Wu
Southwest Jiaotong University
Chengdu, China
wuxiaohk@swjtu.edu.cn

Tianrui Li
Southwest Jiaotong University
Chengdu, China
trli@swjtu.edu.cn

## Abstract

Large pre-trained vision-language models like CLIP have shown amazing zero-shot recognition performance. To adapt pre-trained vision-language models to downstream tasks, recent studies have focused on the "*learnable context + class name*" paradigm, which learns continuous prompt contexts on downstream datasets. In practice, the learned prompt context tends to overfit the base categories and cannot generalize well to novel categories out of the training data. Recent works have also noticed this problem and have proposed several improvements. In this work, we draw a new insight based on empirical analysis, that is, uninformative class names lead to degraded base-to-novel generalization performance in prompt learning, which is usually overlooked by existing works. Under this motivation, we advocate to improve the base-to-novel generalization performance of prompt learning by enhancing the semantic richness of class names. We coin our approach as the **I**nformation **D**isengagement based **A**ssociative **P**rompt **L**earning (IDAPL) mechanism which considers the associative, meanwhile, decoupled learning of prompt context and class name embedding. IDAPL can effectively alleviate the phenomenon of learnable context overfitting to base classes, meanwhile, learning more informative semantic representation of base classes by fine-tuning the class name embedding, leading to improved performance on both base and novel classes. Experimental results on eleven widely used few-shot learning benchmarks clearly validate the effectiveness of our proposed approach. Code is available at https://github.com/tiggers23/IDAPL

## CCS Concepts

• **Computing methodologies** → *Object recognition.*

## Keywords

Vision-language model; prompt learning; cross-category generalization; embedding disengagement

### ACM Reference Format:
Fengmao Lv, Changru Nie, Jianyang Zhang, Guowu Yang, Guosheng Lin, Xiao Wu, and Tianrui Li. 2024. Rethinking the Effect of Uninformative Class Name in Prompt Learning. In *Proceedings of the 32nd ACM International Conference on Multimedia (MM '24), October 28-November 1, 2024, Melbourne, VIC, Australia.* ACM, New York, NY, USA, 10 pages. https://doi.org/10.1145/3664647.3681224

*Corresponding author.

## 1 Introduction

It is well known that the recognition performance of CLIP is usually sensitive to the prompt template. Prompt engineering based on trial and error usually takes a large amount of time in word tuning. To address this problem, existing works [8, 14, 45] assign a set of learnable vectors as the prompt template and optimizes them on downstream datasets. In practice, however, the learned prompt context usually tends to overfit the base categories and cannot generalize to novel categories well. Hence, recent studies along this line mainly focus on improving the cross-category generalizability of prompt learning by learning image conditional prompt [44], multimodal prompt [14] and regularization stratages [15, 46]. To further improve the generalizability of prompt learning, we turn our attention to analyzing the mechanism of cross-category generalizability degradation in prompt learning.

Fengmao Lv, Changru Nie, Jianyang Zhang, Guowu Yang, Guosheng Lin, Xiao Wu, and Tianrui Li

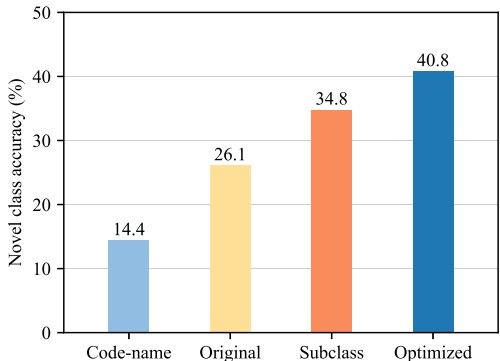

**Figure 1: Novel class accuracies obtained by implementing context optimization on the family-level task of FGVCAircraft dataset [21] with base class names of different qualities, including manually-defined code-names, original name embedding, name embedding enhanced by hierarchical label set [25], and name embedding refined by images [27]. "Code-name" indicates replacing the original base class names with code-names (e.g., "00", "01"). For fair comparisons, the testing procedure is conducted with original name embedding of novel categories.**

To answer the above question, we propose a hypothesis from the perspective of prompt composition, that is, *one of the key reasons for the learned prompts overfitting to base categories is the lack of semantics carried in class name embedding*. The focus of existing works on prompt learning mainly lies in the context part, but ignores that a prompt is composed of a prompt context and a class name (i.e., the "*learnable context + class name*" paradigm). In general, class names are assumed to carry the discriminative information for classification, while the prompt context should be responsible for aligning the text encoder with the corresponding visual features, acting as a universal template. In practice, however, class names are usually coarsely-defined or non-descriptive (e.g., *"A340-200"* representing an aircraft) [23], which may degrade the discriminative performance of CLIP. When the semantic information provided by class names is insufficient, the learnable prompt context will try to derive discriminative information from training data, in order to remedy the uninformative drawback of class names. As a result, the learned context will be biased to base categories and cause performance degradation on novel categories out of the training dataset. This violates the original intention of context optimization, i.e., automatically learning a prompt context acting as a universal template (e.g., "a photo of a"). To verify this hypothesis, we conduct experiments by implementing context optimization with class names of different qualities. From the results shown in Figure 1, we can see that context prompt learned with class names of richer semantic richness (subclass names or optimized class names) can be more generalizable to novel categories. When using manually-defined code-names to replace the class names of training data, the learned context will have very poor recognition performance on novel categories. These experimental phenomena well confirm our hypothesis.

Motivated by the above observations, this work advocates improving the base-to-novel generalization performance of prompt learning by enhancing the semantic richness of class names. To this end, we propose the **A**ssociative **P**rompt **L**earning (APL) mechanism to associatively optimize both the learnable context and class name embedding. Specifically, by introducing learnable class-specific residual vectors, APL sets class name embedding as trainable variables to refine them. During the learning process, the semantic richness of class name embedding can be improved by deriving class-related discriminative information from training data. With trainable class name embedding to remedy the drawback of uninformative class names, the prompt context can hence concentrate on its original role in finding a universal template. As a result, the learned prompt context can achieve improved generalizability on novel categories. In addition, APL can also use class semantic descriptions generated by GPT [26] as additional training data to refine class name embedding following the implementation of [8].

Moreover, although improving the semantic richness of class name embedding can reduce the learnable context overfitting to base classes, there is still no guarantee to prevent learnable contexts from carrying base class information, especially when both context and class name embedding are optimized by a unified cross-entropy loss. To achieve this guarantee, we further propose the **U**niversal and **D**iscriminative **I**nformation **D**isengagement (UDID) mechanism to ensure the prompt context and class name embedding to concentrate on their original role in finding a universal template and carrying the discriminative information of classes, respectively. Specifically, UDID adopts an embedding discriminator to decouple the prompt context and class names. As class name embedding already carries most of the class information, decoupling the prompt context and class names can further prevent it from carrying base class information, as a result, avoiding the learned prompts overfitting to the base classes. We coin our approach as **I**nformation **D**isengagement based **A**ssociative **P**rompt **L**earning (IDAPL) considering both the associative and disengagement mechanism of prompt context and class name embedding. The overall architecture of IDAPL is shown in Figure 2. IDAPL can effectively alleviate the phenomenon of learnable context overfitting to base classes, meanwhile, learning more informative semantic representation of base classes by fine-tuning the class name embedding, leading to improved performance on both base and novel classes. The IDAPL mechanism can be easily applied into existing prompt learning methods (e.g., CoOp [45] and MaPLe [14]). Extensive experiments on diverse few-shot learning benchmarks clearly demonstrate the effectiveness of our approach. Compared with the previous works relying on regularization with human-engineered prompt temples [15, 46], which potentially requires a lot of human costs, IDAPL achieves better performance as well. This result further validates the effectiveness of our approach.

## 2 Related Works

### 2.1 Vision-Language Models

VLMs [13, 29, 40] mainly focus on learning aligned image-language representations. In general, vision-language models consist of a visual encoder and a language encoder and bridge the two modalities by training two encoders jointly. The recent studies on VLMs pay attentions on pre-training large-scale vision-language models via contrastive representation learning [3, 9] on web-scale datasets.

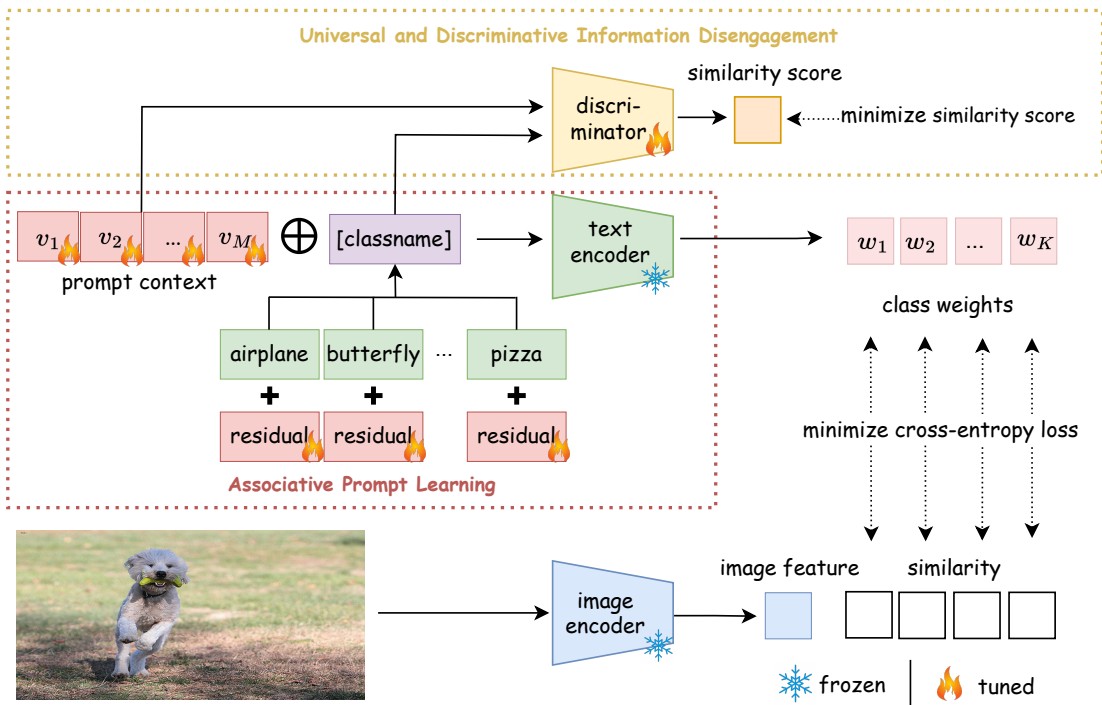

**Figure 2: The overall architecture of Information Disengagement based Associative Prompt Learning (IDAPL). IDAPL consists of two components, the Associative Prompt Learning (APL) mechanism and the Universal and Discriminative Information Disengagement (UDID) mechanism. APL is proposed to learn the ideal prompts by optimizing both prompt context and class names. Specifically, APL learns a general prompt context initialized by "a photo of a" and class-specific residual vectors which are added to original class name embedding. To further prevent the prompt context from carrying base class information, the UDID mechanism decouples the context and class names by minimizing the similarity score of these two embeddings.**

For instance, CLIP [29] and ALIGN [13] respectively exploit about 400 million and 1.8 billion noisy image-text pairs curated from the Internet for pre-training. Based on the rich supervision provided by natural language, large-scale pre-trained VLMs such as CLIP and ALIGN have been recognized to learn powerful visual representations that are transferable across a wide range of downstream tasks.

## 2.2 Prompt Learning

Inspired by the NLP community [17, 18], the prompt learning paradigm is also applied for VMLs adaptation [19, 34–36, 45]. For example, Yu et al. [42] propose to add learnable residuals on classification weights. Zhou et al. [45] propose the CoOp approach to learn continuous prompt contexts on downstream datasets. Yin et al. [41] propose test time prompt tuning to avoid collecting labeled training data for prompt learning. In practice, the learned prompts usually tend to overfit the base categories and cannot well generalize to novel categories out of the training data. Hence, recent works mainly focus on improving the cross-category generalization performance of the prompt learning paradigm [14, 20, 39, 44, 46]. For example, Chen et al. propose to apply optimal transport to match learnable prompt with visual features. Khattak et al. [14] propose MaPLe to improve alignment between the vision and language embedding. Zhou et al. [44] propose the Conditional CoOp (CoCoOp)

approach to improve the cross-category generalizability of prompt learning. Moreover, Maniparambil et al. [46] and Roy et al. [31] propose using GPT generated class description to refine the learnable prompts. Zhu et al. [46] propose the Prompt-aligned Gradient (ProGrad) mechanism to prevent prompt tuning from damaging the prior knowledge of pre-trained VLMs. Khattak et al. [15] propose a self-regularization strategy to improve the cross-category generalizability of learned prompt. However, the above two approaches rely on regularization with human-engineered prompt temples, which potentially require a lot of human costs. In addition,

In general, existing works still lack in-depth analyses of how prompt learning effects the cross-category generalizability of VLMs. In practice, as we found, the uninformative class name embedding could significantly reduce the cross-category generalizability of the learned prompt context. Several recent works have also found the importance of improving the richness of class names for VLMs. Novack et al. [25] and Ge et al. [7] improve the class name embedding via hierarchical label set. Yu et al. [27] further propose the Class name Optimization (CnOp) mechanism to learn optimal class names in prompt learning by replacing the class name embedding with learnable vectors which are trained on downstream tasks. However, the above works lack further insight on the connection between uninformative class names and biased context optimization. In this work, we utilize class name optimization to improve the generalizability of prompt learning in new classes.

## 3 Methodology

In this section, we present our proposed Information Disengagement based Associative Prompt Learning approach, with the motivation of improving the base-to-novel generalizability of prompt learning by enhancing the semantic richness of class names.

### 3.1 Preliminaries

**Revisiting CLIP**. CLIP [29] consists of an image encoder $\mathcal{V}$ and a text encoder $\mathcal{G}$. After pre-trained by a contrastive loss on large-scale multimodal corpus, CLIP can be used for zero-shot classification. For each test image $x$, the image encoder $\mathcal{V}$ encodes it into a visual embedding $\mathbf{f} = \mathcal{V}(x)$. The classification weight vectors $\{\mathbf{w}_i\}_{i=1}^K$ are derived by using hand-engineered text prompts (e.g., "a photo of a [classname]$_i$") to query the textual encoder $\mathcal{G}$, where $K$ is the number of classes and [classname]$_i$ represents the name of class $i$ (e.g., "cat", "dog" and "car"). The prediction probability that image $x$ belongs to class $i$ is calculated as:

$$p(y = i|x) = \frac{\exp\left(\text{sim}\left(\mathbf{f}, \mathbf{w}_i\right)/\tau\right)}{\sum_{j=1}^K \exp\left(\text{sim}\left(\mathbf{f}, \mathbf{w}_j\right)/\tau\right)}, \tag{1}$$

where $\text{sim}(\cdot, \cdot)$ denotes the cosine similarity and $\tau$ is a learned temperature parameter.

**Prompt Learning in CLIP**. As CLIP's performance is sensitive to human-defined prompt context (e.g., "a photo of a"), the context optimization mechanism [45] is proposed to learn the optimal context by introducing a set of learnable vectors $P = \{v_1, v_2, ..., v_M\}$ to replace the human-defined context. The prompts are hence reformulated as $t_i = \{v_1, v_2, ..., v_M, C_i\}$, where $C_i$ represents the word embedding of class names, and $M$ is the number of learnable context vectors. The prediction probability is computed as:

$$p(y = i|x) = \frac{\exp\left(\text{sim}\left(\mathbf{f}, \mathcal{G}(t_i)\right)/\tau\right)}{\sum_{j=1}^K \exp\left(\text{sim}\left(\mathbf{f}, \mathcal{G}(t_j)\right)/\tau\right)}. \tag{2}$$

The prompt context vectors $\{v_1, v_2, ..., v_M\}$ are learned by minimizing the cross-entropy loss between $p(y = i|x)$ and ground-truth labels.

### 3.2 Motivation

In a desired prompt $t_i = \{v_1, v_2, ..., v_M, C_i\}$, the class name embedding should be responsible for carrying the discriminative information for classification, while the prompt context should be responsible for aligning $\mathcal{G}(t_i)$ with the corresponding visual features, acting as a universal template. Existing works put their main focus on optimizing prompt contexts [45], in order to adapt CLIP to downstream datasets, with the assumption that the class name embedding can reveal the visual difference between categories. In practice, however, the class names are usually coarsely-defined or non-descriptive (e.g., *"A340-200"* representing an aircraft). In this case, the context vectors $\{v_1, v_2, ..., v_M\}$ will be trained to derive discriminative information from the training dataset, in order to remedy the uninformative drawback of class names. As a result, the learned prompt context will inevitably be biased towards base categories and cannot generalize well to novel categories out of the training dataset. Hence, the class name embedding should be refined, not only for improving the discriminative ability but also for improving the base-to-novel generalizability of the learned context.

### 3.3 Method

Figure 2 shows the overall architecture of our **I**nformation **D**isengagement based **A**ssociative **P**rompt **L**earning (IDAPL) approach. To achieve the above motivation, IDAPL consists of two components, the **A**ssociative **P**rompt **L**earning (APL) mechanism and the **U**niversal and **D**iscriminative **I**nformation **D**isengagement (UDID) mechanism. APL optimizes both prompt context and class name embedding. With the semantically enriched class name embedding, the generalizability of the learned prompt context can be improved. To further prevent the prompt context from carrying base class information, the UDID mechanism decouples the context and class names by minimizing the similarity of these two embeddings.

**Associative Prompt Learning.** To improve the generalizability of prompt context, we propose the APL mechanism to learn the prompts by optimizing both prompt context and class name embedding. In the "*learnable context + class name*" paradigm, the whole prompt for class $i$ can be defined as $t_i = \{v_1, v_2, ..., v_M, C_i\}$, where $C_i = \{c_{(i,1)}, ..., c_{(i,L_i)}\}$ are the embedding of class name tokens and $L_i$ is the token length of the $i$-th class name. In order to improve the semantic richness of $C_i$, we introduce class-specific category residuals $R_i = \{r_{(i,1)}, ..., r_{(i,L_i)}\}$ to update it as follows:

$$C_i^* = (1 - \alpha) \times C_i + \alpha \times R_i, \tag{3}$$

where each $r_{(i,j)}$ is a learnable vector with the same dimension as the class name embedding $c_{(i,j)}$ and $\alpha$ is a hyper-parameter. By replacing the original class embedding $C_i$ in prompt $t_i$ with $C_i^*$, the refined prompt $t_i^* = \{v_1, v_2, ..., v_M, C_i^*\}$ is received. Thereby, the classification weight vector $\mathbf{w}_i^*$ will be generated by passing the refined prompt $t_i^*$ through the text encoder $\mathcal{G}$ as $\mathbf{w}_i^* = \mathcal{G}(t_i^*)$.

Similar to existing prompt learning approaches [14, 44, 45], we optimize prompts using base class images through the standard classification loss function, cross-entropy loss, formally,

$$\mathcal{L}_I = \frac{1}{|\mathfrak{D}_B|} \sum_{j \in \mathfrak{D}_B} \log\left(p\left(y = I_j|x_j\right)\right), \tag{4}$$

where $\mathfrak{D}_B$ represents the set of base class images, and $I_j$ represents the real label of sample $x_j$. During training, both prompt context vectors (i.e., $\{v_1, v_2, ..., v_M\}$) and category residual variables (i.e, $\{R_i | i \in \mathfrak{C}_B\}$) are optimized by $\mathcal{L}_I$, with the purpose of learning informative class embedding and generalizable prompt context. In addition, our approach can also use class semantic descriptions generated by GPT [26] as additional training data for $\mathfrak{D}_B$ to refine class name embedding by optimizing the Euclidean distance between the feature of GPT descriptions and category prompts.

**Universal and Discriminative Information Disengagement.** Although improving the semantic richness of class name embedding can reduce the learnable context overfitting to base classes, there is still no guarantee to prevent learnable contexts from carrying base class information. To achieve this guarantee, we propose the UDID mechanism, which ensures the prompt context and class name embedding to concentrate on there original role in finding a universal template and carrying the discriminative information of classes, respectively. To achieve this, UDID adopts an embedding discriminator $\mathcal{D}$ to decouples the prompt context $P = \{v_1, v_2, ..., v_M\}$ and refined class names $\{C_1^*, C_2^*, ..., C_K^*\}$. The objective of $\mathcal{D}$ is to learn a binary classifier to classify whether the input embedding is the

general context or class name embedding. This decoupling process is performed using the binary cross-entropy loss, formally,

$$\mathcal{L}_D = -\log\left(\mathcal{D}\left(P\right)\right) - \frac{1}{|\mathfrak{C}_B|} \sum_{i \in \mathfrak{C}_B} \log\left(1 - \mathcal{D}\left(C_i^*\right)\right), \qquad (5)$$

where $\mathfrak{C}_B$ is the set of base classes. In this way, the similarity between prompt context and class names can be reduced. As class name embedding already carries most of the discriminative information, reducing the similarity can prevent it from carrying base class information. Therefore, the disengagement mechanism can further avoid the learned prompt context from overfitting the base classes.

**Optimization and Inference.** The overall training object of our IDAPL is the combination of the above two loss functions, formally,

$$\mathcal{L} = \mathcal{L}_I + \beta \times \mathcal{L}_D, \qquad (6)$$

where $\beta$ is a hyper-parameter. In the inference stage, we put the optimized prompt context $\{v_1, v_2, ..., v_M\}$ and the original novel class name embedding into the text encoder $\mathcal{G}$ to derive the classification weights for novel classes.

Based on the above APL and UDID mechanisms, our IDAPL approach can effectively alleviate the phenomenon of learnable context overfitting to base classes, meanwhile, learning more informative semantic representation of base classes by fine-tuning the class name embedding, leading to improved performance on both base and novel classes.

## 4 Experiments

### 4.1 Experimental Setup

We conduct comprehensive experiments on diverse benchmarks to validate the core idea of our work. Both base-to-novel and cross-dataset generalization settings [14, 44, 45] are considered to demonstrate the effectiveness of our proposed approach in improving the cross-category generalizability of the prompt learning mechanism. Specifically, in the base-to-novel setting, each dataset is divided into subsets of base and novel categories, and only base categories provide images for training. In the cross-dataset generalization setting, we adopt ImageNet [32] for training and test the performance on other datasets with categories different from the training categories.

### 4.2 Dataset

Following existing works on prompt learning [14, 44, 45], we conduct experiments on eleven standard image recognition benchmarks which focus on different categories, including: i) two generic-object recognition benchmarks (i.e., ImageNet [32] and Caltech101 [5]); ii) five fine-grained recognition benchmarks (i.e., OxfordPets [28], StanfordCars [16], Flowers102 [24], Food101 [1], and variant level task of FGVCAircraft [21]); iii) one scene benchmark (i.e., SUN397 [38]); iv) one action recognition benchmark (i.e., UCF101 [33]); v) one texture benchmark (i.e., DTD [4]); vi) one satellite-image benchmark (i.e., EuroSAT [10]).

### 4.3 Implementation Details

In all the experiments, following prior works [14, 44], we randomly select 16 samples per category for training, under the few-shot learning setting. In our experiments, we implement the IDAPL mechanism under the training paradigm of CoOp [45] and MaPLe [14] respectively. Specifically, the pre-trained ViT-B/16 CLIP model is utilized as the backbone of our model. We directly adopt the GPT visual descriptions collected by Maniparambil et al. [22] as the additional training data of base classes. The training epoch number is set to 50 and 5 for CoOp w/ IDAPL and MaPLe w/ IDAPL, respectively. The adaptation stage utilizes the same iteration number as that of the training stage. The hyper-parameter $\alpha$ & $\beta$ are set to 0.1 & 5, respectively. The rest hyper-parameter settings (e.g., learning rate, batch size, context length, prompt depth, etc) follow the original implementation of CoOp [45] and MaPLe [14]. In addition, to warm up the discriminator $\mathcal{D}$, we optimize it by Eq. 4 with the original general context (i.e., "a photo of a") and class name embedding before the formal training process. The hyperparameters are fixed across all datasets. The experiments are conducted on one 4090 GPU. The implementation code will be released online upon acceptance.

### 4.4 Experimental Results

**Baselines**. We compare our approach with the recent state-of-the-art works on VLMs adaptation, including CLIP-Adapter (CLIP-A) [6], Context Optimization (CoOp) [45], Class name Optimization (CnOp) [27], Conditional CoOp (CoCoOp) [44], Prompt Learning with Optimal Transport (CoPLOT) [2], Regularized Mask Tuning(R-AMT) [43], Multi-modal Prompt Learning (MaPLe) [14], CLIP-A-Self [22], PromptSRC [15], and ProGrad [46]. Of these, CLIP-A [6], R-AMT [43] and CoOp [44] are pioneer works on adapting CLIP towards downstream tasks. Specifically, CLIP-A introduces lightweight additional feature adapters on either visual or language branches, R-AMT fine-tunes VLMs via masking selected parameters, while CoOp focuses on learning continuous context vectors. CnOp [27] proposes to adapt CLIP towards downstream datasets by optimizing class name embedding. CoCoOp [44], CoPLOT [2], MaPLe [14], PromptSRC [15], and ProGrad [46] draw attentions on improving the base-to-novel generalization performance of prompt learning. CLIP-A-Self [22] further proposes to enhance prompt learning by utilizing GPT descriptions.

**Base-to-Novel Generalization**. This setting splits datasets into base and novel categories. In this setting, the model is first trained on base categories and then tested on novel categories. Table 1 displays the performance comparison of each approach. From Table 1, we can draw the following observations. First, the performance achieved by our approach is on par with or even better than that of the compared baselines on most of the benchmarks (except FGV-CAircraft, Caltech101, and ImageNet), as well as on average. By improving the generalizability of prompt context and enriching the class name embedding together, our approach can obtain consistent performance improvement over both base and novel categories. Second, by applying the IDAPL mechanism into the training paradigm of CoOp, our approach can achieve consistent performance improvement over the vanilla CoOp approach. By applying the

**Table 1: Comparison with existing prompt learning methods on the base-to-novel generalization setting. H indicates the harmonic mean of accuracies on base and novel categories.**

(a) Average over 11 datasets

|  | Base | Novel | H |
|---|---|---|---|
| CLIP | 69.3 | 74.2 | 71.7 |
| CLIP-A | 79.9 | 72.3 | 75.9 |
| CoOp | 82.7 | 63.2 | 71.7 |
| CoCoOp | 80.5 | 71.7 | 75.8 |
| CoPLOT | 75.9 | 67.6 | 71.8 |
| CnOp | 82.8 | 73.1 | 77.7 |
| R-AMT | **85.7** | 72.2 | 78.4 |
| MaPLe | 82.3 | 75.1 | 78.5 |
| CLIP-A-Self | 82.5 | 74.5 | 78.3 |
| PromptSRC | 84.3 | 76.1 | 80.0 |
| ProGrad | 82.5 | 70.7 | 76.2 |
| CoOp+IDAPL | 83.8 | 75.1 | 79.2 |
| MaPLe+IDAPL | 84.2 | **77.7** | **80.8** |

(b) ImageNet

|  | Base | Novel | H |
|---|---|---|---|
| CLIP | 72.4 | 68.1 | 70.2 |
| CLIP-A | 75.4 | 68.6 | 71.8 |
| CoOp | 76.5 | 67.9 | 71.9 |
| CoCoOp | 76.0 | 70.4 | 73.1 |
| CoPLOT | 68.2 | 63.1 | 65.7 |
| CnOp | 74.7 | 68.1 | 71.2 |
| R-AMT | 77.2 | 70.3 | 73.6 |
| MaPLe | 76.7 | 70.5 | 73.5 |
| CLIP-A-Self | 76.4 | 68.3 | 72.1 |
| PromptSRC | **77.6** | 70.7 | **74.0** |
| ProGrad | 77.0 | 66.7 | 71.5 |
| CoOp+IDAPL | 77.1 | 70.6 | 73.7 |
| MaPLe+IDAPL | 76.8 | **70.8** | 73.7 |

(c) FGVCAircraft

|  | Base | Novel | H |
|---|---|---|---|
| CLIP | 27.2 | 36.3 | 31.1 |
| CLIP-A | 34.9 | 33.5 | 34.2 |
| CoOp | 40.4 | 22.3 | 28.7 |
| CoCoOp | 33.4 | 23.7 | 27.7 |
| CoPLOT | 25.6 | 26.6 | 26.1 |
| CnOp | 44.0 | 33.3 | 37.9 |
| R-AMT | **49.2** | 32.1 | 38.9 |
| MaPLe | 37.4 | 35.6 | 36.5 |
| CLIP-A-Self | 37.8 | 33.0 | 35.2 |
| PromptSRC | 42.7 | **37.9** | **40.2** |
| ProGrad | 40.5 | 27.6 | 32.8 |
| CoOp+IDAPL | 44.8 | 34.2 | 38.8 |
| MaPLe+IDAPL | 44.0 | 35.9 | 39.6 |

(d) Food101

|  | Base | Novel | H |
|---|---|---|---|
| CLIP | 90.1 | 91.2 | 90.7 |
| CLIP-A | 90.3 | 91.2 | 90.7 |
| CoOp | 88.3 | 82.3 | 85.2 |
| CoCoOp | 90.7 | 91.3 | 91.0 |
| CoPLOT | 85.0 | 85.2 | 85.1 |
| CnOp | 86.5 | 91.0 | 88.7 |
| R-AMT | 90.7 | 91.1 | 90.9 |
| MaPLe | 90.7 | 92.1 | 91.4 |
| CLIP-A-Self | 90.4 | 91.2 | 90.8 |
| PromptSRC | 90.7 | 91.5 | 91.1 |
| ProGrad | 90.4 | 89.6 | 90.0 |
| CoOp+IDAPL | 90.3 | 91.7 | 91.0 |
| MaPLe+IDAPL | **90.9** | **92.1** | **91.5** |

(e) Caltech101

|  | Base | Novel | H |
|---|---|---|---|
| CLIP | 96.8 | 94.0 | 95.4 |
| CLIP-A | 97.7 | 93.6 | 95.6 |
| CoOp | 98.0 | 89.8 | 93.7 |
| CoCoOp | 98.0 | 93.8 | 95.8 |
| CoPLOT | 95.4 | 90.9 | 93.2 |
| CnOp | 97.9 | 94.0 | 95.9 |
| R-AMT | **98.9** | 94.4 | 96.6 |
| MaPLe | 97.7 | 94.4 | 96.0 |
| CLIP-A-Self | 98.3 | **95.9** | **97.1** |
| PromptSRC | 98.1 | 94.0 | 96.0 |
| ProGrad | 98.0 | 93.9 | 95.9 |
| CoOp+IDAPL | 98.6 | 95.2 | 96.9 |
| MaPLe+IDAPL | 97.8 | 94.2 | 96.0 |

(f) Flowers102

|  | Base | Novel | H |
|---|---|---|---|
| CLIP | 72.1 | 77.8 | 74.8 |
| CLIP-A | 94.6 | 71.5 | 81.4 |
| CoOp | 97.6 | 59.7 | 74.1 |
| CoCoOp | 94.9 | 71.8 | 81.7 |
| CoPLOT | 89.6 | 69.2 | 79.4 |
| CnOp | 98.0 | 77.1 | 86.3 |
| R-AMT | 98.0 | 70.9 | 82.3 |
| MaPLe | 95.9 | 72.5 | 82.6 |
| CLIP-A-Self | 97.4 | 75.3 | 84.9 |
| PromptSRC | **98.1** | 76.5 | 86.0 |
| ProGrad | 95.5 | 71.9 | 82 |
| CoOp+IDAPL | 97.7 | **78.1** | **86.8** |
| MaPLe+IDAPL | 96.7 | 76.4 | 85.3 |

(g) OxfordPets

|  | Base | Novel | H |
|---|---|---|---|
| CLIP | 91.2 | 97.3 | 94.1 |
| CLIP-A | 94.8 | 97.0 | 95.9 |
| CoOp | 93.7 | 95.3 | 94.5 |
| CoCoOp | 95.2 | 97.7 | 96.4 |
| CoPLOT | 92.1 | 95.9 | 94.0 |
| CnOp | 93.4 | 96.9 | 95.1 |
| R-AMT | **95.7** | 96.0 | 95.8 |
| MaPLe | 95.4 | **97.8** | 96.6 |
| CLIP-A-Self | 94.4 | 97.0 | 95.7 |
| PromptSRC | 95.3 | 97.3 | 96.3 |
| ProGrad | 95.1 | 97.6 | 76.2 |
| CoOp+IDAPL | 95.4 | 97.3 | 96.4 |
| MaPLe+IDAPL | 95.5 | 97.7 | **96.6** |

(h) StanfordCars

|  | Base | Novel | H |
|---|---|---|---|
| CLIP | 63.4 | 74.9 | 68.7 |
| CLIP-A | 70.5 | 73.3 | 71.9 |
| CoOp | 78.1 | 60.4 | 68.1 |
| CoCoOp | 70.5 | 73.6 | 72.0 |
| CoPLOT | 63.2 | 66.5 | 64.9 |
| CnOp | 80.6 | 75.0 | 77.7 |
| R-AMT | **82.9** | 69.5 | 75.6 |
| MaPLe | 72.9 | 74.0 | 73.5 |
| CLIP-A-Self | 76.8 | 72.9 | 74.8 |
| PromptSRC | 78.3 | 75.0 | 76.6 |
| ProGrad | 77.7 | 68.6 | 72.9 |
| CoOp+IDAPL | 81.4 | 75.3 | **78.3** |
| MaPLe+IDAPL | 80.8 | **75.5** | 78.1 |

(i) SUN397

|  | Base | Novel | H |
|---|---|---|---|
| CLIP | 69.4 | 75.4 | 72.2 |
| CLIP-A | 80.1 | 75.9 | 77.9 |
| CoOp | 80.6 | 65.9 | 72.5 |
| CoCoOp | 79.7 | 76.9 | 78.3 |
| CoPLOT | 75.2 | 73.2 | 74.2 |
| CnOp | 79.1 | 75.5 | 77.3 |
| R-AMT | 82.2 | 76.5 | 79.2 |
| MaPLe | 80.8 | 78.7 | 79.7 |
| CLIP-A-Self | 81.4 | 76.8 | 79.0 |
| PromptSRC | **82.7** | 78.5 | 80.5 |
| ProGrad | 81.3 | 74.2 | 77.6 |
| CoOp+IDAPL | 82.2 | 78.1 | 80.1 |
| MaPLe+IDAPL | 81.9 | **79.5** | **80.7** |

(j) DTD

|  | Base | Novel | H |
|---|---|---|---|
| CLIP | 53.2 | 59.9 | 56.4 |
| CLIP-A | 74.9 | 53.0 | 62.1 |
| CoOp | 79.4 | 41.2 | 54.2 |
| CoCoOp | 77.0 | 56.0 | 64.9 |
| CoPLOT | 72.6 | 51.4 | 62.0 |
| CnOp | 80.3 | 60.0 | 68.7 |
| R-AMT | **84.4** | 57.2 | 68.2 |
| MaPLe | 80.4 | 59.2 | 68.2 |
| CLIP-A-Self | 81.8 | 62.3 | 70.7 |
| PromptSRC | 83.4 | 63.0 | 71.7 |
| ProGrad | 77.4 | 52.4 | 62.4 |
| CoOp+IDAPL | 82.2 | 57.1 | 67.4 |
| MaPLe+IDAPL | 83.4 | **65.3** | **73.3** |

(k) EuroSAT

|  | Base | Novel | H |
|---|---|---|---|
| CLIP | 56.5 | 64.1 | 60.0 |
| CLIP-A | 82.5 | 62.4 | 71.1 |
| CoOp | 92.2 | 54.7 | 68.7 |
| CoCoOp | 87.5 | 60.0 | 71.2 |
| CoPLOT | 91.0 | 55.3 | 73.2 |
| CnOp | 92.1 | 59.8 | 72.5 |
| R-AMT | **95.8** | 58.3 | 72.5 |
| MaPLe | 94.1 | 73.2 | 82.4 |
| CLIP-A-Self | 88.5 | 70.5 | 78.5 |
| PromptSRC | 92.9 | 73.9 | 82.3 |
| ProGrad | 90.1 | 60.9 | 72.7 |
| CoOp+IDAPL | 86.9 | 73.2 | 79.5 |
| MaPLe+IDAPL | 93.9 | **85.8** | **89.7** |

(l) UCF101

|  | Base | Novel | H |
|---|---|---|---|
| CLIP | 70.5 | 77.5 | 73.9 |
| CLIP-A | 82.9 | 74.9 | 78.7 |
| CoOp | 84.7 | 56.1 | 67.5 |
| CoCoOp | 82.3 | 73.5 | 77.6 |
| CoPLOT | 77.4 | 66.2 | 71.8 |
| CnOp | 84.1 | 73.9 | 78.7 |
| R-AMT | **87.9** | 77.4 | 82.3 |
| MaPLe | 83.0 | 78.7 | 80.8 |
| CLIP-A-Self | 84.1 | 76.4 | 80.1 |
| PromptSRC | 87.1 | 78.8 | 82.7 |
| ProGrad | 84.3 | 74.9 | 79.4 |
| CoOp+IDAPL | 84.7 | 74.8 | 79.4 |
| MaPLe+IDAPL | 84.4 | **81.4** | **82.9** |

**Table 2: Performance comparison with the existing methods of prompt learning under the cross-dataset generalization setting.**

| | Source | Target | | | | | | | | | | |
|---|---|---|---|---|---|---|---|---|---|---|---|---|
| | ImageNet | Flowers102 | Aircraft | Food101 | Caltech101 | OxfordPets | StanfordCars | SUN397 | DTD | EuroSAT | UCF101 | Average |
| CoOp | 71.5 | 68.7 | 18.5 | 85.3 | 93.7 | 89.1 | 64.5 | 64.2 | 41.9 | **46.4** | 66.6 | 63.9 |
| CoCoOp | 71.0 | 71.9 | 22.9 | 86.1 | **94.4** | 90.1 | 65.3 | 67.4 | 45.7 | 45.4 | 68.2 | 65.7 |
| MaPLe | 71.6 | 70.8 | 21.8 | 86.1 | 92.1 | 90.1 | 65.6 | 66.5 | 46.7 | 35.3 | 67.8 | 64.3 |
| PromptSRC | 71.3 | 70.3 | **23.9** | 86.2 | 93.6 | **90.3** | **65.7** | 67.1 | 46.9 | 45.5 | 68.8 | **65.8** |
| ProGrad | 72.2 | 67.9 | 20.6 | 85.4 | 91.5 | 89.6 | 62.4 | 62.5 | 39.4 | 43.5 | 64.3 | 62.7 |
| CoOp+IDAPL | **72.5** | 71.7 | 23.1 | **86.4** | **94.4** | 89.8 | 65.6 | 67.2 | **47.0** | 39.5 | **69.1** | 65.4 |
| MaPLe+IDAPL | 72.1 | **72.6** | 23.8 | 86.0 | 94.0 | 90.1 | 65.6 | **68.0** | 45.9 | 38.7 | 68.9 | 65.4 |

**Table 3: Ablation study on base-to-novel generation task under the training paradigm of CoOp+IDAPL. We report the average results of all the 11 datasets.**

| | Base | Novel |
|---|---|---|
| "a photo of a" | 69.3 | 73.1 |
| + context learning (CoOp) | 82.1 | 68.8 |
| + associative learning | 83.5 | 71.9 |
| + GPT generated description | 83.8 | 74.7 |
| + UDID (full model) | **83.8** | **75.1** |

**Table 4: Performance on the domain generalization setting, in which models are trained on ImageNet and then evaluated on datasets of other domain styles.**

| | Source | Target | | | | |
|---|---|---|---|---|---|---|
| | ImageNet | -V2 | -S | -A | -R | Avg. |
| CoOp | 71.5 | 64.2 | 48.0 | 49.7 | 75.2 | 59.3 |
| CoOp+IDAPL | **72.6** | **65.1** | 48.7 | 50.4 | 76.1 | 60.1 |
| MaPLe | 70.7 | 64.1 | **49.2** | **50.9** | **77.0** | **60.3** |
| MaPLe+IDAPL | 70.9 | 64.0 | **49.2** | 50.7 | 76.9 | 60.2 |

**Table 5: Comparison for the number of trainable parameters on the Flowers102 dataset.**

| Method | Parameters | | Performances | | |
|---|---|---|---|---|---|
| | Num. | % CLIP | Base | Novel | H |
| CoOp | 2048 | 0.002 | 97.6 | 59.7 | 74.1 |
| CoOp+IDAPL | 134.7 K | 0.108 | 97.5 | 78.4 | 86.9 |
| MaPLe | 3.55 M | 2.85 | 95.9 | 72.5 | 82.6 |
| MaPLe+IDAPL | 3.68 M | 2.96 | 97.7 | 77.5 | 86.5 |

IDAPL mechanism into the training paradigm of MaPLe, our approach outperforms the vanilla MaPLe approach on most of the benchmarks (except Caltech101), as well as on average. In addition, our approach can clearly outperform the CLIP-A-Self baseline which also utilizes GPT descriptions to enhance prompt learning. Finally, our approach can clearly outperform the regularization strategy based baselines, PromptSRC [15] and ProGrad [46], which potentially require a lot of human costs. This result shows the superiority of enhancing the semantic richness of class names and decoupling the embedding of prompt context and class names in improving the generalizability of prompt learning.

**Cross-Dataset Generalization**. In this setting, the model is first trained on base categories of one source dataset and then tested on novel categories of other datasets. Table 2 displays the performance comparison of each approach. From Table 2, we can draw similar observations as in the above setting. In general, our approach can achieve improved performance compared with baseline models and can achieve comparable (3/10) or improved (6/10) performance compared with other comparison models on target benchmarks (except EuroSAT). EuroSAT is a satellite-image benchmark, which is very different from other datasets and contains only 10 classes, making its performance unstable. Fair comparisons, we adopt the same hyperparameters for baseline models, (i.e., CoOp and MaPLe) and our implementation.

## 4.5 Analysis

**Ablation Study**. In order to further demonstrate the effectiveness of the proposed associative learning mechanism, we conduct ablation studies and show the corresponding results in Table 3. Comparing CLIP with the context learning mechanism, we can see that learning prompt context can significantly improve base class accuracy, but limit the cross-category generalizability as the

learned context overfits to base classes. By introducing associative learning of both prompt context and class name embedding, we can observe consistent performance improvements on both base and novel classes. In addition, using class semantic descriptions generated by GPT as additional training data to refine the prompts also achieves performance improvement. These observations show that the APL mechanism can alleviate the phenomenon of the learned context overfit to base classes, meanwhile, learning more informative semantic representation of base classes by fine-tuning the class name embedding with semantic rich class descriptions and base class images, which is consistent with our motivation. Furthermore, comparing the fourth line with the fifth line in Table 3, it is clear that our UDID mechanism achieves performance improvement on novel classes, showing that decoupling the learnable context and class name embedding could further protect the learnable context from overfitting to base classes.

**Sensitivity Analysis**. To demonstrate the robustness of our approach, we further conduct sensitivity analysis for hyper-parameters and report the corresponding results in Figure 3. From the upper part of Figure 3, we can see that the performance of our approach progressively improves as the epoch number grows, and is stabilized after 50 epochs, while the performance of CoOp is sensitive

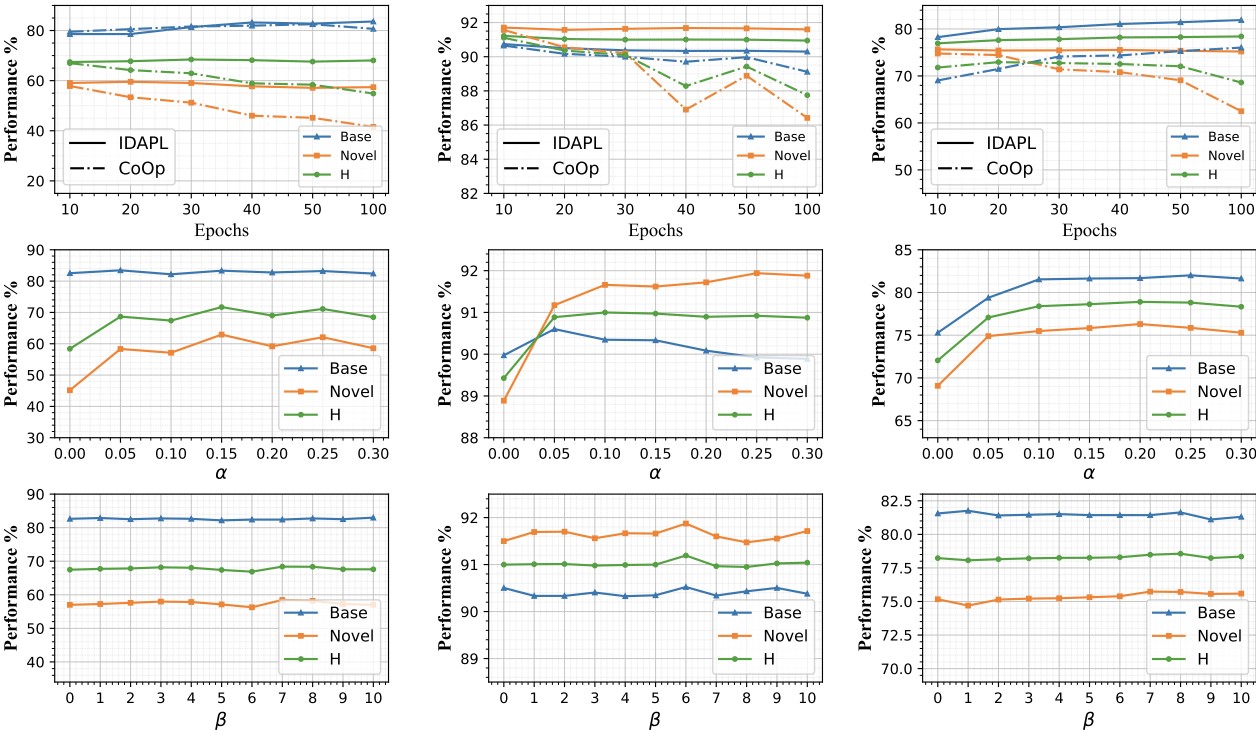

**Figure 3: Sensitivity analysis on DTD (*left*), StanfordCars (*middle*), and Food101 (*right*) datasets. The results are obtained by varying the value of the corresponding hyper-parameter, while fixing the other hyper-parameters to the values adopted in the experiments.**

to the epoch number. This observation further shows that our proposed associative learning mechanism is robust against base-to-novel performance degradation during context optimization. From the middle and bottom part of Figure 3, we can see that our approach can tolerate wide ranges of the trade-off parameters $\alpha$ & $\beta$, which are used to incorporate category residuals $R_i$ into class name embedding and balance the loss functions, respectively.

**Domain Generalization**. The associative prompt learning mechanism is originally proposed for improving the cross-category generalization performance of context optimization. Considering that our approach usually obtains better performance on base categories, we have concerns on whether the introduced IDAPL mechanism will cause extra bias towards the domain style of training data. To this end, we further examine the performance of our approach on the domain generalization setting and report the corresponding results in Table 4. Specifically, the model is first trained on ImageNet and then evaluated on datasets of other domain styles (i.e., ImageNet-V2 [30], ImageNet-S [37], ImageNet-A [12] and ImageNet-R [11]). From Table 4, we can see that the introduced IDAPL mechanism does not lead to clear performance degradation on new domains. The domain generalization performance obtained by our approach can be on par with or even better than the corresponding baselines, although cross-domain generalization is not the original purpose of our approach.

**Prompt Complexity**. As shown in Table 5, we compare the number of trainable parameters of CoOp+IDAPL and MaPLe+IDAPL with their baselines. We can see that the IDAPL mechanism only

introduces about 0.1M extra trainable parameters (∼0.1% parameters of CLIP) to their baselines, but brings about clear performance improvement on the harmonic mean metrics. Notably, since APL does not change the visual branch, its inference speed is consistent with the original baseline model.

## 5 Conclusion

This work focuses on improving the generalization ability of prompt learning for vision-language models. To achieve this, we draw a novel empirical-analysis-based insight on the effect of uninformative class names causing novel class performance degradation in prompt learning. Motivated by this observation, we propose our IDAPL approach, which consists of APL and UDID mechanisms. APL alleviates the phenomenon of learnable context overfitting to base classes by enhancing the semantic richness of class names, meanwhile, semantically enriched class name embedding also brings performance improvement on base classes. In addition, as APL cannot prevent the learnable context from learning base class information, we further propose UDID to ensure the prompt context and class name embedding to concentrate on their original role in finding a universal template and carrying the discriminative information of classes, respectively, by decoupling the embedding of prompt context and class names. In this way, the learnable context is protected from biasing to base classes. We conduct extensive experiments on eleven widely used few-shot learning benchmarks. Experimental results clearly validate the effectiveness of IDAPL.

## Acknowledgments

This work was supported by the National Natural Science Foundation of China (No. 62106204, 62176221, 61572407), the Fundamental Research Funds for the Central Universities (No. 2682024ZTPY055), the Sichuan Science and Technology Program (No. 2024NSFTD0036, 2024ZHCG0166 ), the Frontier Cross Innovation Team Project of Southwest Jiaotong University (YH1500112432297), the CSC scholarship, and the Engineering Research Center of Sustainable Urban Intelligent Transportation, Ministry of Education.

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
