# OpenReview forum: "Rethinking the Effect of Uninformative Class Name in Prompt Learning"
_acmmm.org/ACMMM/2024/Conference — MM2024 Poster_

### Official Review · Reviewer_6Rbt · 2024-05-21

**Rating:** 5
**Confidence:** 3

**Summary:**

This paper introduces the Information Disengagement based Associative Prompt Learning (IDAPL) mechanism to enhance the base-to-novel generalization performance of prompt learning by enriching the semantic content of class names. Experimental results on eleven few-shot learning benchmarks demonstrate that IDAPL improves performance on both base and novel classes.

**Strengths:**

The motivation of this paper is very clear. This paper optimize the context prompt with class names of  semantic richness to generalize pretrained model to novel categories.

The paper is well-organized and well-written.

The experiments are thorough and validate the effectiveness of the method.

**Limitations:**

The author proposes deriving class-related discriminative information from training data by learning residuals. Could this objective also be achieved by directly learning class names? If optimization is difficult, could the corresponding learning rate be adjusted?

Ablation studies were only conducted on 11 fine-grained recognition datasets. Verification of the method’s effectiveness is also needed on ImageNet variant datasets.

In line 25, “meanwhile” is confusing.

By applying the proposed IDAPL to the training paradigm of MaPLe and CoOp, the performance of vanilla MaPLe and CoOp has improved on most benchmarks. However, similar results were not observed on Caltech101. The authors should provide some explanations. Additionally, on most datasets, the proposed method does not outperform PromptSRC in terms of base performance, which also requires an explanation.

**Suitability:**

2

---

### Official Review · Reviewer_VCuD · 2024-05-25

**Rating:** 3
**Confidence:** 3

**Summary:**

This paper proposes to enhance the semantic richness of class names to improve the base-to-novel generalization performance of prompt learning. Experimental results validate the effectiveness of our proposed approach.

**Strengths:**

1. The paper is well-written and easy to follow.
2. The analysis in Lines 102-170 is clear and insightful.
2. The proposed UDID mechanism is interesting.

**Limitations:**

1. APL can assist categories with insufficient information such as "A340-200", but such categories only make up a small portion. For other conventional categories, it may already have sufficient information to distinguish them (comparing their category embedding similarity may indicate this), and the performance benefits mainly come from the additional information provided by APL and GPT, which may conflict with the motivation of the paper.
2. Based on the first point of view, I speculate that the main improvement comes from the additional information brought by GPT, rather than APL. I am curious about the performance of removing the APL module and only using the GPT-generated description.
3. The proposed strategy does not bring benefits to Domain Generalization, which is somewhat puzzling. I hope the author can discuss this more.

**Suitability:**

2

---

### Official Review · Reviewer_B7q6 · 2024-05-26

**Rating:** 4
**Confidence:** 3

**Summary:**

This paper investigates the impact of uninformative class names on the generalization performance of prompt learning. The study reveals that the lack of semantically rich class names leads to prompt context overfitting to base categories, resulting in poor performance on novel categories. To address this issue, the authors propose the Information Disengagement based Associative Prompt Learning (IDAPL) method, which enhances the generalizability of prompts by optimizing both the prompt context and class name embeddings. IDAPL consists of two main mechanisms: Associative Prompt Learning (APL) and Universal and Discriminative Information Disengagement (UDID). APL improves class name embeddings by introducing class-specific residual vectors, while UDID uses an embedding discriminator to decouple the prompt context from class name embeddings. Experimental results demonstrate that IDAPL outperforms existing methods on various few-shot learning benchmarks, improving classification performance on both base and novel categories.

**Strengths:**

1. The proposed method is simple yet effective.
2. The paper is well writen and easy to understand.
3. The overall idea is novel.

**Limitations:**

1. During inference stage, is the  class-specific category residuals used for base class? Will this cause inconsistent inference model for base and novel class? For more realistic settings such as evaluation among all classes[a], the proposed method might be invalid.
2. The details of GPT generated class semantic descriptions are not provided.



[a]. Shu Y, Guo X, Wu J, et al. Clipood: Generalizing clip to out-of-distributions[C]//International Conference on Machine Learning. PMLR, 2023: 31716-31731.

**Suitability:**

3

---

### Official Review · Reviewer_uw3w · 2024-05-26

**Rating:** 2
**Confidence:** 4

**Summary:**

This paper introduce a new method which improves the base-to-novel generalization performance of prompt learning by enhancing the semantic richness of class names.

**Strengths:**

IDAPL can be combined with multiple SOTA methods (CoOP and MaPLe) without introducing much learnable params.

**Limitations:**

The improvement on performance is limited according Table1/2. Moreover, according to Table.3, CoOp+IDAPL uses gpt generated prompts. Is the comparison reported in Table.1 fair enough? Because the results of CoCoOp in this paper is same as the results reported by CoCoOp itself, which did not use gpt generated prompts.

More SOTAs ([1][2][3) should be included for comparison in Table.1 and Fig.3.

What is APL in Fig.3? I think it might be IDAPL.

[1]PLOT: Prompt Learning with Optimal Transport for Vision-Language Models (ICLR23)
[2]Regularized Mask Tuning: Uncovering Hidden Knowledge in Pre-trained Vision-Language Models (ICCV23)
[3]Task Residual: Tuning Vision-Language Models in One Line of Code (CVPR23)

**Suitability:**

3

---

### Meta-Review · Area_Chair_3K2g · 2024-06-25

**Recommendation:** Accept (Poster)
**Confidence:** 5

**Metareview:**

This paper examines the impact of uninformative class names on the generalization performance of prompt learning. The authors find that semantically poor class names cause the prompt context to overfit to base categories, resulting in poor performance on novel categories. To address this, they propose the Information Disengagement based Associative Prompt Learning (IDAPL) method, which optimizes both prompt context and class name embeddings to enhance generalizability. IDAPL includes two main components: Associative Prompt Learning (APL), which introduces class-specific residual vectors to improve class name embeddings, and Universal and Discriminative Information Disengagement (UDID), which uses an embedding discriminator to separate the prompt context from class name embeddings. Experimental results show that IDAPL outperforms existing methods on various few-shot learning benchmarks, improving classification performance on both base and novel categories.

The proposed method is simple yet effective. The paper is well-written and easy to understand, presenting a novel idea that marks a significant advancement in the field.

More discussions with respect to recent methods is desried:
[1]PLOT: Prompt Learning with Optimal Transport for Vision-Language Models (ICLR23)
[2]Regularized Mask Tuning: Uncovering Hidden Knowledge in Pre-trained Vision-Language Models (ICCV23)
[3]Task Residual: Tuning Vision-Language Models in One Line of Code (CVPR23)
[4] In-context Prompt Learning for Test-time Vision Recognition with Frozen Vision-language Model. CoRR abs/2403.06126 (2024)